# Response of Evapotranspiration, Photosynthetic Characteristics and Yield of Soybeans to Groundwater Depth

Zhenchuang Zhu [1] , Zhijun Chen [2], Zhe Wang [1], Ruxuan Shen [1] and Shijun Sun [1,*]

1   College of Water Conservancy, Shenyang Agricultural University, Shenyang 110866, China; 2020200012@stu.syau.edu.cn (Z.Z.)
2   China Agricultural Water Research Center, China Agricultural University, Beijing 100083, China
*   Correspondence: sunshijun2000@syau.edu.cn

**Abstract:** To clarify the physiological mechanism of different groundwater depths affecting soybean evapotranspiration, photosynthetic characteristics and yield, a field experiment with four groundwater depth levels (1 m (D1), 2 m (D2), 3 m (D3) and 4 m (D4)) was conducted through the groundwater simulation system in 2021 and 2022. In this study, a quantitative analysis was conducted on the groundwater recharge and irrigation water demand and evapotranspiration (ET) of soybean fields with different treatments, and the effects of different treatments on soybean leaf area index (LAI), chlorophyll content index (SPAD), intercepted photosynthetic active radiation (IPAR), photosynthetic gas exchange parameters, dry matter accumulation (DMA) and yield were explored. The results showed the following: (1) Groundwater depth affected soybean ET and the source of ET. With the increase in groundwater depth, groundwater recharge and its contribution to ET gradually decreased, but the amount of irrigation required gradually increased, resulting in the ET as D1 > D4 > D2 > D3. (2) Soybean LAI, SPAD and IPAR were significantly affected by the different groundwater depths, of which the D1 treatment always maintained the maximum, followed by the D4 treatment, and the D3 treatment was the minimum. The photosynthetic gas exchange parameters under different treatments changed synergistically, showing significant differences in the flowering and podding stages, notably D1 > D4 > D2 > D3. Soybean DMA and yield first decreased and then increased with the increase in groundwater depth, and the average DMA and yield under the D1 treatment increased by 27.71%, 46.80% and 22.82% and 20.29%, 29.91% and 12.83% in the two years, respectively, compared to the D2, D3 and D4 treatments. (3) The structural equation model demonstrated that the groundwater depth indirectly affected the growth of soybean leaf area by affecting groundwater recharge, which in turn regulated soybean ET and photosynthetic capacity and ultimately affected DMA and yield. The above results showed that in the case of shallow groundwater depth (D1), the largest groundwater recharge promoted the growth of soybean leaf area and chlorophyll synthesis and increased the absorption and utilization of solar radiation. And it improved the leaf stomata conditions, accelerated the gas exchange between the plant and atmosphere, enhanced the photosynthetic production capacity and ET and achieved maximum DMA and yield. Soybean leaf growth and photosynthesis diminish with the increase in groundwater depth. In the case of deep groundwater depth (D4), the maximum irrigation improved the growth and photosynthetic performance of soybean leaves, which was favorable to ET, and ultimately led to increases in DMA and yield.

**Keywords:** groundwater depth; soybean; groundwater recharge; evapotranspiration; intercepted photosynthetic active radiation; photosynthetic characteristics

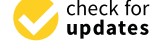



## 1. Introduction

In shallow groundwater areas, a full farming water system consists of groundwater, soil moisture, plant water and atmospheric water. Soil moisture and groundwater are intimately related to one another and interact with each other. Meanwhile, soil moisture status and crop growth interact and constrain each other [1,2]. Soil moisture status causes

diverse responses in crops that affect everything from morphology to physiology, so the presence of shallow groundwater will inevitably have an impact on the process of crop growth and evapotranspiration (ET) [3]. In recent years, many scholars have conducted extensive research on the impact of shallow groundwater on crops. The soil capillary action provided more groundwater to the crop root zone, thereby increasing soil moisture content [4]. And as the groundwater depth increased, the capillary effect in the soil weakened. When the groundwater level was too deep, the rising zone of capillary water could not reach the soil layer of crop roots, which led to drought stress in the crop root zone and was unfavorable for crop growth and ET [5,6]. Kang et al. [7] found that the effect of a groundwater depth of 2 m on crop growth was greater than that of a groundwater depth of 3 and 4 m, which showed that the larger the groundwater depth, the more unfavorable to the plant height, leaf area index (LAI) and dry matter accumulation (DMA) of winter wheat. She et al. [8] concluded that compared to groundwater depths of 3 and 4 m, the groundwater depth of 2 m helped to extend the rapid growth time and increase the LAI of maize. Wang [9] controlled the groundwater depth at 1, 2, 3 and 4 m and found that shallow groundwater helped to accelerate the reproductive process of plants and increase the LAI, which in turn enhanced plant photosynthesis and ET. Nevertheless, when groundwater depth was too shallow, it also led to poor soil ventilation and low oxygen content, resulting in the obstruction of root growth, the early decline in the upper functional leaves of the plants, a reduction in leaf area and chlorophyll and decreased photosynthetic capacity, ultimately reducing crop ET and yield [10,11]. In addition, shallow groundwater depth increases the risk of secondary soil salinization because of the presence of higher levels of salts in groundwater or soil [12]. Obviously, groundwater depth is an important environmental factor affecting crop growth and development. It is worth noting that different crops have different requirements for groundwater depth [6,13,14]. Previous studies on the effects of groundwater depth on crops have focused on some crops such as wheat, maize and cotton [3,7,8,15], while little research has been reported on the physiological growth response of soybeans to groundwater depth.

Soybean is native to China and has been cultivated and consumed for about 5000 years [16]. With a protein content of about 40% and a vegetable oil content of about 20% [17], soybeans are an important source of vegetable protein and edible oils in people's daily diets. In the last century, China has transformed from the largest soybean-producing country to the largest soybean-importing country. In 2021, total global soybean production reached 368 million tons, while Chinese soybean production accounted for only 5.3% of global production [18]. Over 80% of soybean consumption relies on imports to meet its huge domestic demand [19]. To improve this situation and, as such, increase soybean production, China subsequently issued a series of soybean stimulus policies, such as the soybean revitalization plan in 2019 [20]. The Chinese No. 1 central documents emphasize twice the need to vigorously implement the soybean and oilseed production capacity improvement project in 2022 and 2023 [21].

Photosynthesis and ET are important physiological processes that determine the growth and development status of crops and are also major factors in improving crop productivity. As a light-loving crop, soybeans accumulate 91.31% of the dry matter generated by photosynthesis products [22]. Present studies on the response of soybean ET, photosynthetic characteristics and yield to groundwater depth are still unclear. Therefore, taking soybeans as the research object, different groundwater depths were set based on automatic control systems. In this study, we aimed (1) to quantify the amount of groundwater recharge, required irrigation water and ET in soybean fields at different groundwater depth treatments; (2) to investigate the response of soybean canopy leaf area growth and development, intercepted photosynthetic active radiation, photosynthetic gas exchange parameters, DMA and yield to different groundwater depth treatments; (3) to reveal the influence mechanism of groundwater depth on soybean DMA and yield based on a structural equation model. Our findings can provide theoretical value for guaranteeing stable

and high regional soybean yields and a green and efficient use of groundwater resources in shallow groundwater areas.

## 2. Materials and Methods

### 2.1. Site Description

Field experiments were performed in the dry field test pit area at the experimental station of Liaoning Irrigation Experimental Center Station (42°09′ N, 120°31′ E, 47 m.s.l) in 2021 and 2022. The test pits are fully enclosed bottomed pits of reinforced concrete structure with a maximum depth of 5 m and an area of 5 m² (2 m × 2.5 m). The study area is characterized by a temperate continental monsoon climate with an average annual temperature of 8.5 °C, an average annual precipitation of 699.1 mm and an annual sunshine duration of 2439.4 h. The soil of the study site is a powdery loam with a pH of 7.28. In the 0–20 cm soil layer, the topsoil contains 21.6 g·kg$^{-1}$ organic matter, 1.08 g·kg$^{-1}$ total N, 23.1 mg·kg$^{-1}$ available P and 142.5 mg·kg$^{-1}$ available K.

### 2.2. Research Program

#### 2.2.1. Experimental Design

Four groundwater depth levels (1 m (D1), 2 m (D2), 3 m (D3) and 4 m (D4)) were set by the automatic groundwater depth control system. The experiment was laid out as a randomized block group design, with each treatment being replicated three times. The automatic groundwater depth control system (Figure 1) is mainly composed of water tanks, water level holders, water columns (0.23 m in diameter), test pits and connectors. The water columns and test pits are connected by electromagnetic valves to form a communicating vessel. The groundwater depths in the test pits meet the test requirements by setting the water heights of the water columns, and the electromagnetic valves control the water inflow and outflow in real time to maintain system stability. A mobile rain shelter is set up in the test area to isolate natural precipitation.

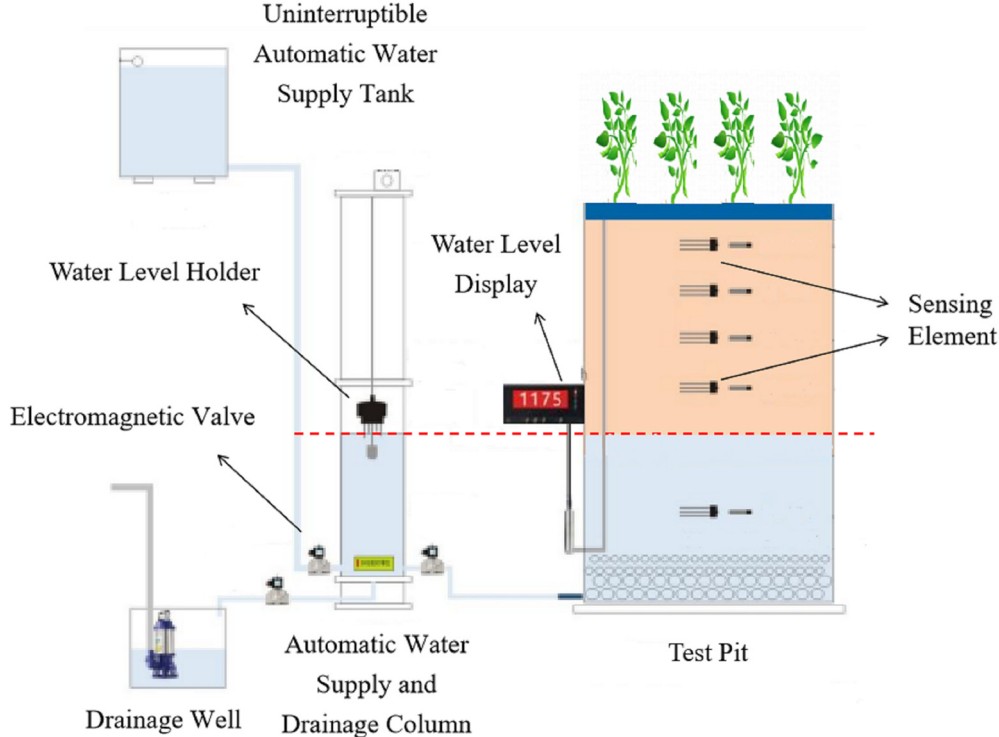

**Figure 1.** Schematic diagram of automatic control system for groundwater depth.

### 2.2.2. Crop Management and Irrigation

A soybean variety (cv. Tiefeng 31) was used with a traditional large-ridge double-line planting method (Figure 2). The sowing density is 180,000 plants·ha$^{-1}$. A total of 150 kg·ha$^{-1}$ of bottom fertilizer (Diammonium phosphate, N-P$_2$O$_5$ 18–46%) was applied at the same time as sowing, and 20 mm of irrigation was applied after sowing to ensure seedling emergence. Soybean was planted on 13 May 2021 and 12 May 2022, then harvested on 4 October 2021 and 27 September 2022. The irrigation method was shallow-buried drip irrigation, and drip irrigation belts were laid between soybean rows. The water used for controlling groundwater depth and irrigation was derived from local groundwater with a mineralization of 0.24 g·L$^{-1}$ and an electrical conductivity of 114 µS·cm$^{-1}$. The amount of irrigation water for soybeans was determined using a calculation of the upper and lower soil moisture content limits (Table 1) [9]. The field capacity was determined using the ring knife method [23], with a value of 33.6% (percentage of volume). Timely weed control was implemented to avoid the influence of grass on the study. Experimental plots are shown in Figure 3.

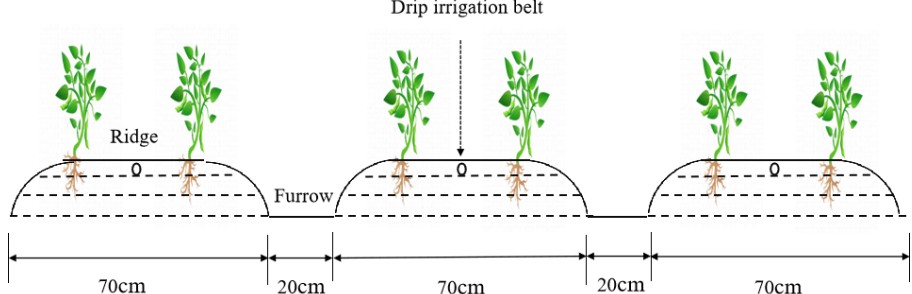

**Figure 2.** Schematic diagram of soybean planting mode.

**Table 1.** Upper and lower limits of soil moisture control (percentage of field capacity). The planned depth of the moist layer is 0–20 cm at the seedling stage, 0–40 cm at the branching stage and 0–50 cm at other growth stages. No irrigation water supply at the maturity stage of soybeans.

| Growth Stage | Lower Limits of Soil Moisture (%) | Upper Limits of Soil Moisture (%) |
|---|---|---|
| Seedling stage | 65 | 75 |
| Branching stage | 70 | 80 |
| Flowering stage | 75 | 85 |
| Podding stage | 75 | 85 |
| Pod-filling stage | 65 | 75 |

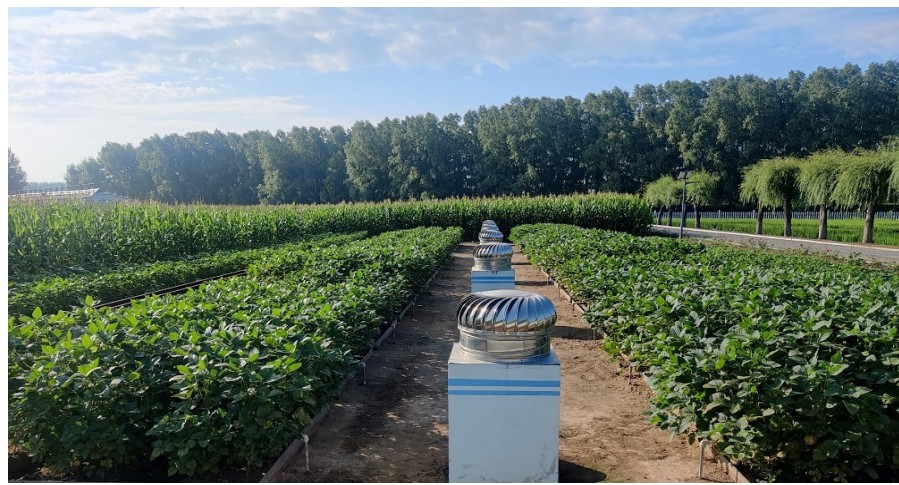

**Figure 3.** Field photograph of the experimental plots.

*2.3. Measurements*

2.3.1. Groundwater Recharge and Irrigation

The automatic groundwater depth control system was used to automatically record the inflow and outflow of the test pits.

$$\text{The amount of groundwater recharge (mm) = inflow of the test pit (mm)} - \text{outflow of the test pit (mm)} \tag{1}$$

The soil moisture sensors (Bolun Jingwei Technology Development Co., Ltd., Beijing, China) were used to automatically record the soil volumetric moisture content of different soil layers. When soil moisture dropped to the lower limit of soil moisture control, the amount of irrigation was determined by referring to the calculation method of Wang et al. [24]:

$$I = 10 \cdot p \cdot H \cdot (\beta_1 - \beta_2) \cdot FC, \tag{2}$$

where I is the amount of irrigation (mm); p is the percentage of soil wetting volume, 70%; H is the planned depth of the moist layer (cm); $\beta_1$ is the upper limit of soil moisture control; $\beta_2$ is the actual soil moisture after the soil moisture dropped to the lower limit of soil moisture control and FC is the soil field capacity.

2.3.2. Evapotranspiration (ET) and Groundwater Contribution

Referring to the calculation method of Huang et al. [25], the ET of soybeans was calculated using the water balance equation:

$$ET = P + I + G - D - R + SWD, \tag{3}$$

where P is the effective precipitation (mm); G is the upward recharge of groundwater (mm); D is the deep soil leakage (mm); R is the soil surface runoff (mm) and SWD is the soil water consumption during the entire growth period of soybeans (mm). Due to the fact that this experiment was conducted under a rain shelter and each plot was a separate measurement pit, P and R were ignored. The experiment was conducted using drip irrigation, which was not prone to deep seepage, so D was neglected. The value of G was obtained from Equation (1). The SWD was calculated as follows [26]:

$$SWD = SWS_{end} - SWS_{begin}, \tag{4}$$

$$SWS = \sum_{i=0}^{n} \theta_i \cdot \Delta z_i, \tag{5}$$

where $SWS_{end}$ and $SWS_{begin}$ represent the soil water storage at the end and beginning of the soybean growth period, respectively (mm); $\theta_i$ is the corresponding volumetric water content in the soil layer i ($cm^3 \cdot cm^{-3}$) and $\Delta z_i$ is the corresponding soil thickness for the layer i (mm).

The contribution rate of groundwater to ET is the ratio of groundwater recharge amount to soybean ET [9].

2.3.3. Leaf Area Index (LAI)

The perforation weighing method was adopted to determine the leaf area of soybeans. Three uniform-growth plants were selected from each treatment. The middle leaves of the functional leaves were taken, avoiding the veins, and ten small discs (6 mm in diameter) were removed with a perforator. The fresh weight of ten small discs and all the leaves in the plant were taken. Referring to the calculation method of Pacheco et al. [27], the LAI was calculated as follows:

$$\text{Leaf area (m}^2\text{) = area of ten small discs (m}^2\text{)} \times \text{fresh weight of all the leaves (g)/fresh weight of ten small discs (g)} \tag{6}$$

$$\text{Leaf area index (LAI) = leaf area per plant (m}^2 \cdot \text{plant}^{-1}\text{)} \times \text{number of plants per unit area (plant} \cdot \text{m}^{-2}\text{)} \tag{7}$$

### 2.3.4. Chlorophyll Relative Content (SPAD)

We selected five uniform-growth plants from each plot and measured the SPAD of the middle leaves of soybean functional leaves using a SPAD-502 handheld chlorophyll meter (Likaile Technology Development Co., Ltd., Beijing, China).

### 2.3.5. Intercepted Photosynthetic Active Radiation (IPAR)

The PAR was measured using an AccuPAR LP-80 canopy analyzer (Ligaotai Technology Co., Ltd., Beijing, China) from 9:00 to 11:00 a.m. on typical sunny days. The PAR was measured at 10 cm above the canopy and on the surface of the soil, respectively. Referring to the slightly revised calculation method of Liu et al. [28], the IPAR was calculated as follows:

$$
\begin{aligned}
\text{Intercepted photosynthetic active radiation (IPAR, } \mu mol \cdot m^{-2} \cdot s^{-1}) = \text{photosynthetic} \\
\text{active radiation (PAR) at 10 cm above the canopy } (\mu mol \cdot m^{-2} \cdot s^{-1}) - \text{photosynthetic} \\
\text{active radiation (PAR) on the surface of the soil } (\mu mol \cdot m^{-2} \cdot s^{-1})
\end{aligned}
\tag{8}
$$

### 2.3.6. Photosynthetic Gas Exchange Parameters

The photosynthetic gas exchange parameters were measured using an LCpro-SD photosynthesizer (Ligaotai Technology Co., Ltd., Beijing, China) from 9:00 to 11:00 a.m. on typical sunny days. The net photosynthetic rate (Pn), stomatal conductance (Gs), intercellular $CO_2$ concentration (Ci) and transpiration rate (Tr) of the middle leaves of the functional soybean leaves were measured and recorded.

### 2.3.7. Dry Matter Accumulation (DMA) and Yield

After the soybean matures, three uniform-growth plants were selected from each treatment, and complete crowns were obtained by cutting off the base of the stems. Samples were taken separately from the stems, leaves and pods, and they were placed in an oven for 30 min at 105 °C and then dried to a constant weight at 80 °C. The weights were determined with a balance (accuracy of 0.01 g). The DMA is the total weight of the stems, leaves and pods. Plot plants were harvested in their entirety and weighed after threshing and natural air drying to derive plot yields, and yields were calculated at 14% moisture content.

### 2.4. Statistical Analysis

DPS 7.05 (Hangzhou Ruifeng Information Technology Co., Ltd., Hangzhou, China) and Origin 2016 (OriginLab Corporation, Northampton, MA, USA) were used for data analysis and graph plotting. Amos 17.0 (IBM SPSS, Chicago, IL, USA) was adopted to perform structural equation analysis. Differences were considered statistically significant at $p < 0.05$ according to the least significant difference (LSD).

## 3. Results

### 3.1. Groundwater Recharge and Irrigation

Different groundwater depths had significant impacts on groundwater recharge and irrigation water demand for soybeans (Figure 4). The amount of groundwater recharge gradually decreased with increasing groundwater depth. Compared with the D1 treatment, the amount of groundwater recharge under the D2, D3 and D4 treatments decreased by 81.10%, 96.81% and 97.45% and 80.65%, 96.69% and 97.25%, respectively, in 2021 and 2022. On the contrary, the amount of irrigation water needed increased with increasing groundwater depth. The groundwater in the D1 treatment could meet the water needs of soybeans throughout the whole growth period, but irrigation was needed at the sowing date (20 mm). The amount of irrigation water needed for the D1 treatment was only 91.73%, 92.99% and 94.18% and 90.89%, 92.88% and 93.95% of that needed for the D2, D3 and D4 treatments in 2021 and 2022, respectively.

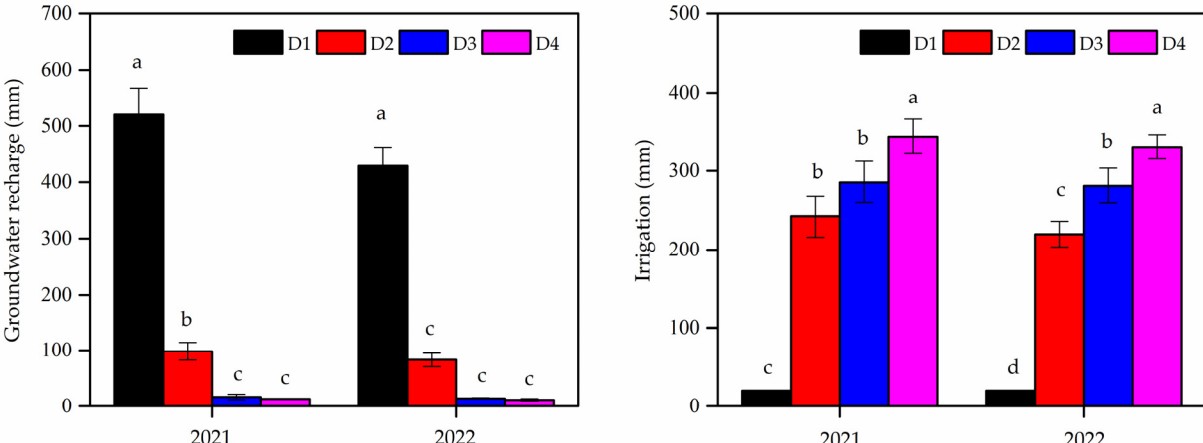

**Figure 4.** Effects of different groundwater depths on the amount of groundwater recharge and irrigation for soybeans in 2021 and 2022. D1, D2, D3 and D4 indicate the groundwater depths of 1, 2, 3 and 4 m, respectively. Different lowercase letters indicate significant differences between different treatments ($p < 0.05$).

### 3.2. Evapotranspiration (ET) and Groundwater Contribution

Different groundwater depths had a significant impact on soybean ET during the entire growth period (Figure 5). Compared to the D1 treatment, the ET decreased significantly with the increase in groundwater depth, with the minimum ET value observed in the D3 treatment. Compared with the ET of the D1 treatment, the ET of D2, D3 and D4 treatments was 56.09%, 79.20% and 41.17% lower, respectively, in 2021 and 43.18%, 52.34% and 24.41% lower, respectively, in 2022. The groundwater contribution to ET tended to decrease with increasing groundwater depth, with the greatest decrease observed between the D1 and D2 treatments, and the groundwater contribution tended to be 0 for the D4 treatment.

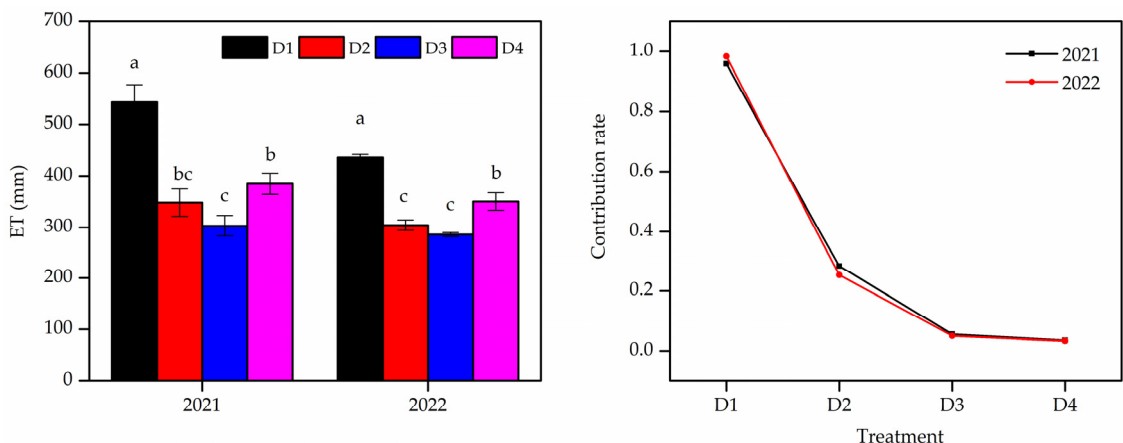

**Figure 5.** Effects of different groundwater depths on the soybean ET and the contribution rate of groundwater to ET in 2021 and 2022. D1, D2, D3 and D4 indicate the groundwater depths of 1, 2, 3 and 4 m, respectively. Different lowercase letters indicate significant differences between different treatments ($p < 0.05$).

### 3.3. Leaf Area Index (LAI)

There were significant differences in the LAI of soybeans under different treatments (Table 2). Under each treatment, the LAI showed a trend with the advancement of the growth process, displaying a single peak parabolic change that first increased and then decreased, reaching its peak at the podding stage of soybean. The two-year data showed that the order of the LAI with different treatments was as follows: D1 > D4 > D2 > D3, i.e., with the increase in groundwater depth, the LAI first decreased and then increased. The

minimum value of the LAI was found in the D3 treatment. At the podding stage, when the LAI reached its peak, the LAI under the D1 treatment increased by 16.22% compared to the D3 treatment in 2021 and increased by 19.29% and 24.88% compared to the D2 and D3 treatments in 2022, respectively.

**Table 2.** Effects of different groundwater depths on LAI of soybeans in 2021 and 2022. D1, D2, D3 and D4 indicate the groundwater depths of 1, 2, 3 and 4 m, respectively. Different lowercase letters indicate significant differences between different treatments ($p < 0.05$).

| Year | Treatment | Branching Stage | Flowering Stage | Podding Stage | Pod-Filling Stage |
|------|-----------|-----------------|-----------------|---------------|-------------------|
| 2021 | D1 | 3.40 ± 0.37 a | 5.05 ± 0.34 a | 5.66 ± 0.39 a | 4.98 ± 0.23 a |
|      | D2 | 2.46 ± 0.25 b | 4.00 ± 0.41 bc | 5.15 ± 0.33 ab | 4.63 ± 0.36 ab |
|      | D3 | 2.39 ± 0.29 b | 3.91 ± 0.31 c | 4.87 ± 0.29 b | 4.32 ± 0.26 b |
|      | D4 | 2.89 ± 0.30 ab | 4.44 ± 0.19 b | 5.30 ± 0.21 ab | 4.67 ± 0.32 ab |
| 2022 | D1 | 2.93 ± 0.43 a | 4.81 ± 0.31 a | 5.07 ± 0.42 a | 4.44 ± 0.10 a |
|      | D2 | 2.20 ± 0.27 b | 3.80 ± 0.34 b | 4.25 ± 0.28 b | 3.78 ± 0.19 b |
|      | D3 | 2.17 ± 0.25 b | 3.71 ± 0.44 b | 4.06 ± 0.33 b | 3.60 ± 0.24 b |
|      | D4 | 2.38 ± 0.36 ab | 4.25 ± 0.31 ab | 4.42 ± 0.28 ab | 3.83 ± 0.28 b |

### 3.4. Chlorophyll Content Index (SPAD)

Different groundwater depths had a significant impact on the SPAD of soybean leaves (Figure 6). With the advancement of the growth process, the SPAD under different treatments gradually increased, reaching a peak at 90–100 DAS, after which the chlorophyll gradually decomposed and the SPAD began to decrease. In 2021, during the whole growth period of soybeans, the SPAD remained the maximum in the D1 treatment, followed by the D4 treatment, and the SPAD under the D3 treatment obtained the minimum value. In 2022, the SPAD of the D1 treatment remained the maximum, too. And before reaching the peak of SPAD, the SPAD of the D2 treatment was greater than that of the D3 treatment, but with little difference. However, the SPAD of the D2 treatment was comparable to the D4 treatment after reaching the peak of SPAD.

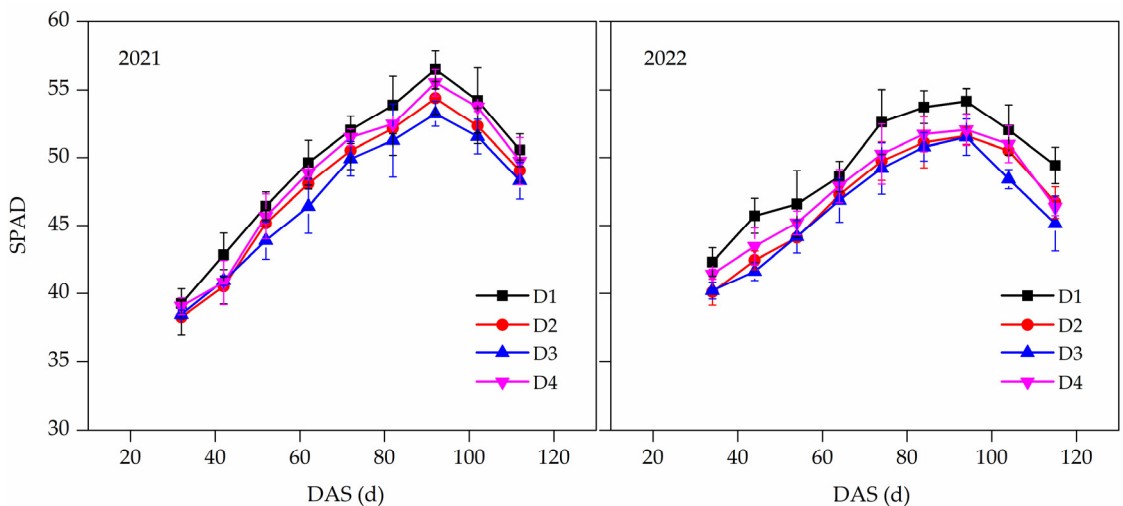

**Figure 6.** Effects of different groundwater depths on SPAD of soybean leaves in 2021 and 2022. D1, D2, D3 and D4 indicate the groundwater depths of 1, 2, 3 and 4 m, respectively.

### 3.5. Intercepted Photosynthetic Active Radiation (IPAR)

Different groundwater depths had a significant impact on the IPAR of the soybean population (Figure 7). The trends of IPAR under different treatments with the advancement of the growth process were consistent. It gradually increased during the early stage of soybean growth, reached a peak value at 80–90 DAS and then gradually decreased due to

the yellowing and shedding of the lower leaves of the plants. In 2021, during the whole soybean growth season, the IPAR under different treatments followed the following order: D1 > D4 > D2 > D3. In 2022, The IPAR of the D1 treatment always obtained the maximum value, followed by the D4 treatment. The IPAR of the D2 treatment was slightly larger than that of the D3 treatment at 60–100 DAS, while the SPAD between the two treatments was almost consistent during the remaining growth periods.

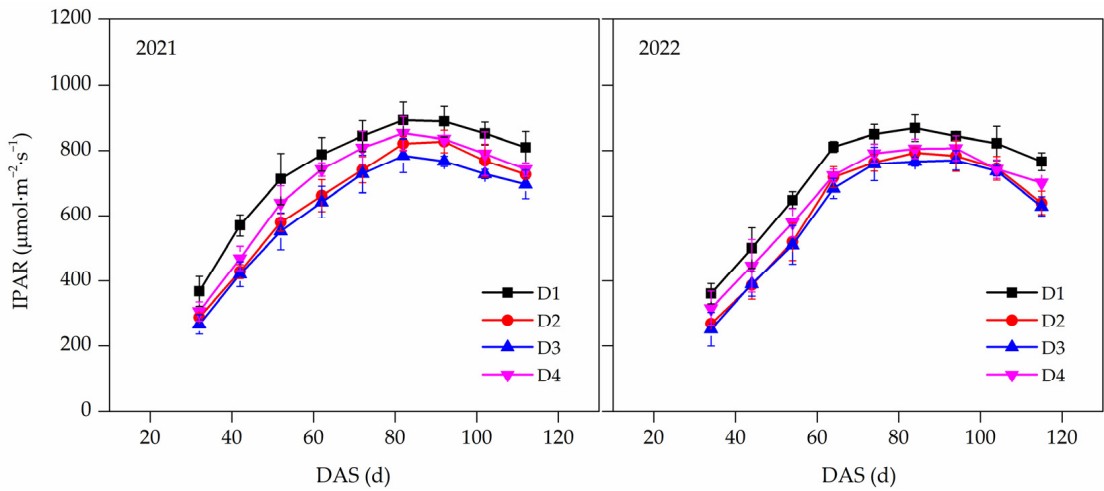

**Figure 7.** Effects of different groundwater depths on IPAR of soybean population in 2021 and 2022. D1, D2, D3 and D4 indicate the groundwater depths of 1, 2, 3 and 4 m, respectively.

*3.6. Photosynthetic Gas Exchange Parameters*

ANOVA revealed significant effects of the different treatments on photosynthetic gas exchange parameters in 2021 and 2022 (Figure 8). With the advancement of the growth process, Pn, Gs, Ci and Tr all first increased under different treatments, reaching peak values at the flowering stage of soybeans, and then began to decrease, which was jointly determined by atmospheric environmental conditions and soybean growth status. At the branching stage, there were no significant differences in photosynthetic parameters under different treatments except for Gs in 2022, which increased by 32.88% and 24.66% in the D1 and D4 treatments, respectively, compared to the D3 treatment. At the flowering and podding stages, the photosynthetic parameters showed significant differences under different treatments, with the D1 treatment obtaining the maximum value, followed by the D4 treatment, and the D3 treatment obtaining the minimum value. At the pod-filling stage in 2021, the Gs under the D1 treatment increased by 33.58%, 38.59% and 28.59% compared to the D2, D3 and D4 treatments, respectively; furthermore, the Tr under the D1 treatment increased by 12.71% and 13.21% compared to the D2 and D3 treatments. At the pod-filling stage in 2022, the Ci under the D1 and D4 treatments increased by 18.44% and 17.93% compared to the D3 treatment, respectively; moreover, the Tr under the D1 treatment increased by 36.26%, 60.67% and 28.83% compared to the D2, D3 and D4 treatments, respectively.

*3.7. Dry Matter Accumulation (DMA) and Yield*

The soybean DMA in different treatments first decreased and then increased with increasing groundwater depth, while it was the lowest for the D3 treatment (Figure 9). The DMA of the D1 treatment was significantly larger than that of the other three treatments and increased by 23.18%, 45.85% and 19.44% and 32.24%, 47.75% and 26.20% in two years compared to the D2, D3 and D4 treatments, respectively. The DMA under the D2 and D4 treatments increased by 18.38% and 22.08% in 2021 and increased by 11.73% and 17.08% in 2022 compared to the D3 treatment. From the viewpoint of different reproductive organs of soybean plants, the proportion of DMA in pods was the largest, while the proportion of DMA in leaves was the smallest. The DMA in pods and stems under different

groundwater depths showed a trend of D1 > D4 > D2 > D3. Different from that, the DMA of soybean leaves showed a gradual decrease with the increase in groundwater depth, i.e., D1 > D2 > D3 > D4. Figure 8 also showed that the trends of soybean yields and DMA under different treatments were consistent, with D1 > D4 > D2 > D3. Among the four treatments, the yield of the D1 treatment was the largest, which increased by 18.04%, 27.52% and 13.08% in 2021 and by 22.54%, 32.29% and 12.58% in 2022 compared to the D2, D3 and D4 treatments, respectively.

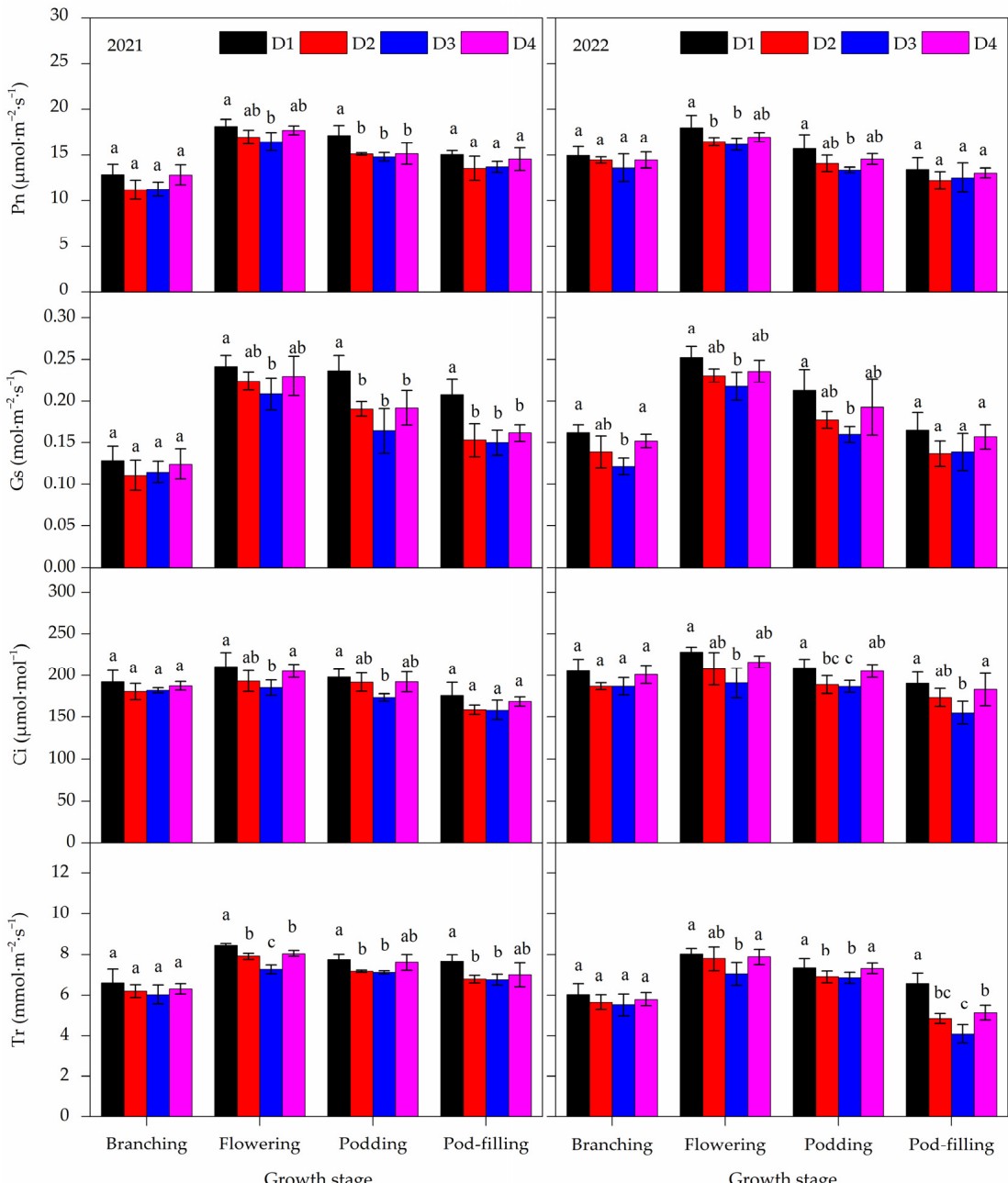

**Figure 8.** Effects of different groundwater depths on photosynthetic gas exchange parameters of soybean in 2021 and 2022. D1, D2, D3 and D4 indicate the groundwater depths of 1, 2, 3 and 4 m, respectively. Pn, Gs, Ci and Tr indicate net photosynthetic rate, stomatal conductance, intercellular $CO_2$ concentration and transpiration rate, respectively. Different lowercase letters indicate significant differences between different treatments ($p < 0.05$).

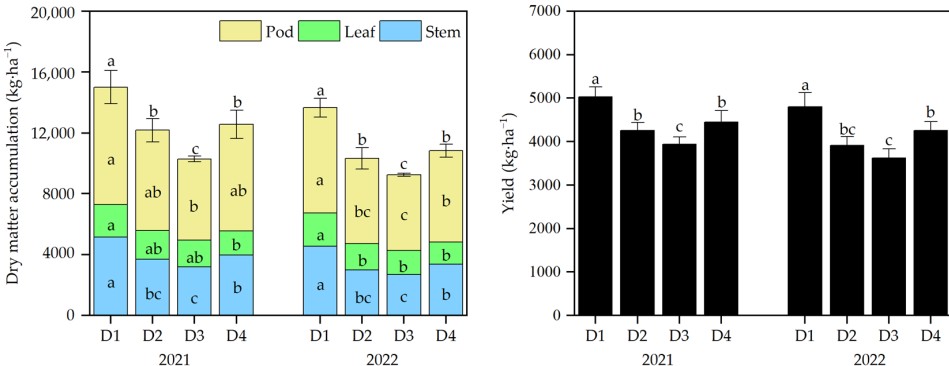

**Figure 9.** Effects of different groundwater depths on DMA and yield of soybean in 2021 and 2022. D1, D2, D3 and D4 indicate the groundwater depths of 1, 2, 3 and 4 m, respectively. Different lowercase letters indicate significant differences between different treatments ($p < 0.05$).

### 3.8. Structural Equation Modeling Analysis

A structural equation model (SEM) based on the causal links associated with various indexes was created, taking into account that soybean yield was the outcome of numerous indexes working together. The possible paths of all indexes were included in the SEM, and the output of the model results are shown in Figure 10. The model validation indicated that the model fit well with the observed data. As illustrated in Figure 8, the groundwater depth had a negative direct effect on groundwater recharge and a positive direct effect on irrigation water demand. The groundwater depth had different degrees of indirect effects on the LAI, SPAD, IPAR and ET through groundwater recharge. The LAI had positive effects on ET and Tr. The IPAR indirectly affected DMA through Tr, Gs and Ci of soybean photosynthetic parameters, thereby affecting soybean yield. In addition, the IPAR also affected yield through Pn. The results indicated that the groundwater depth directly affected the amount of groundwater recharge, indirectly affected the growth and development of soybean leaf area and thereby regulated soybean ET and photosynthetic capacity, ultimately affecting DMA and yield.

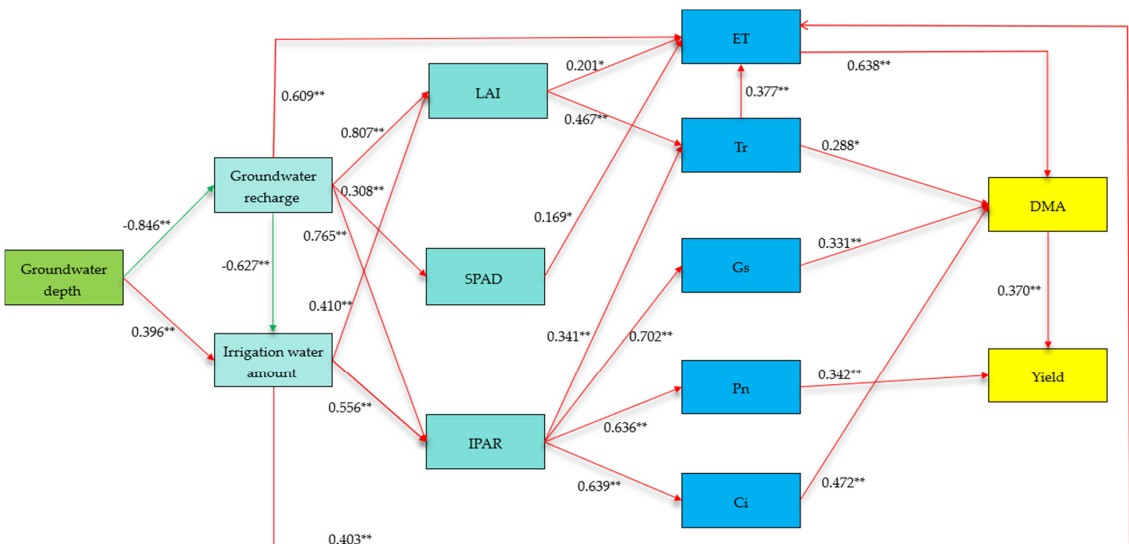

**Figure 10.** SEM (structural equation model) fitted to connections of different indexes (CMIN/DF = 2.411, GFI = 0.958, CFI = 0.961, RMSEA = 0.048, P = 0.163). LAI: leaf area index; SPAD: chlorophyll content index; IPAR: intercepted photosynthetic active radiation; ET: evapotranspiration; Pn: net photosynthetic rate; Gs: stomatal conductance; Ci: intercellular $CO_2$ concentration; Tr: transpiration rate; DMA: dry matter accumulation. Numbers adjacent to arrows represent the standardized path coefficients (r) (** $p < 0.01$, * $p < 0.05$).

## 4. Discussion

In shallow groundwater zones, the upward movement of groundwater under capillary force plays an important role in supplementing unsaturated soil water, especially in the presence of crops. The water absorption effect of crop roots greatly accelerates the intensity of groundwater movement towards unsaturated soil water and improves soil water conditions [4]. The height and amount of capillary water rise directly affect the growth status of crops. Huo et al. [29] established a one-dimensional variable saturated flow model using Hydrus-1D software (version 3.0) and simulated it under stable groundwater depth and continuous increases in groundwater depth, respectively. The simulation results indicated that the increase in the thickness of the unsaturated zone delayed the recharge process of groundwater, and as the groundwater depth increased, the recharge amount significantly decreased. In this study, we found that the groundwater recharge was intense under the D1 treatment, resulting in almost no need for irrigation throughout the entire growth period of soybeans, and relying solely on groundwater recharge could meet the needs of soybean growth and development. In this case, the ET was the maximum, and the groundwater contribution to ET was also the maximum. As groundwater recharge decreased with increasing groundwater depth, even with supplementary irrigation water, soil moisture conditions remained near the lower limit of soil moisture control for an increasingly long period of time, resulting in a decreasing ET of soybean, and the contribution of groundwater to ET gradually decreased. When the groundwater depth was 4 m (D4), the capillary rising water struggled to affect the root soil layer, and the soil moisture was always rapidly consumed to the lower limit of control. Then, irrigation was carried out in time, resulting in the soil moisture condition being better than the groundwater depth of 2 and 3 m for most of the time, which promoted the increase in soybean ET. In this situation, groundwater contributed the least to ET. It can be seen that the effect of groundwater depth on soybean ET is not only limited to the amount of ET but also includes the differences in ET sources.

Leaves are the main organs for photosynthesis in crops, and the LAI is an important index reflecting the crop's growth status and photosynthetic production capacity. Setiyono et al. [30] concluded that soybean LAI demonstrated a trend of first increasing and then decreasing with the advancement of the growth period, which is consistent with the conclusion drawn in our study. Our study also revealed that the LAI of the D1 treatment remained at the maximum, with the D4 treatment coming in second and the D3 treatment at the minimum (Table 2). It did, however, deviate slightly from the research findings of Kang et al. [7], and further investigation revealed that all groundwater depth treatments in their study used the same amount of irrigation water. The SPAD is usually used to reflect the chlorophyll content of plant leaves, which is closely related to crop photosynthetic efficiency and DMA [31]. In this study, the SPAD also revealed D1 > D4 > D2 > D3 at different groundwater depths (Figure 6). The primary cause is that, under the D1 treatment, the intense capillary action of the soil supplies the soybean root zone with more groundwater, which encourages the increase in the soybean leaf area and the production of SPAD. The amount of groundwater recharge was reduced with the increase in the groundwater depth. Furthermore, because of the abnormally deep groundwater under the D4 treatment, the groundwater recharge's contribution to soybean growth was minimal. In this case, the maximum amount of irrigation also promoted the growth of the soybean leaf area, which is in line with the findings of Al-ghawry et al. [32]. PAR is the component of solar radiation that can be used by crops for photosynthesis [28]. We discovered that soybean IPAR varied in concert with LAI and SPAD under different groundwater depth treatments. It can be seen that the amount of PAR absorbed by plant leaves, in addition to being affected by changes in solar radiation, is also related to the structure of the plant canopy. The growth and development status of the canopy leaves is the core factor affecting the utilization of light energy and the photosynthetic production capacity of crops [33].

Previous studies have shown that the two main aspects affecting plant photosynthesis under environmental stress conditions include stomatal and non-stomatal limitation [34,35]. If Pn and Ci fluctuate in the same direction, it suggests that stomatal restriction is the

primary factor affecting photosynthetic rate. The decrease in stomatal conductance leads to a decrease in $CO_2$ content entering the stomata of leaves and makes it impossible for plants to maintain high-intensity photosynthesis. The most critical difference should be in the chloroplasts, which is caused by non-stomatal factors, if the direction of change for Ci and Pn is the reverse [35,36]. Gas exchange parameters such as Pn, Gs, Ci and Tr followed the pattern D1 > D4 > D2 > D3 under different groundwater depth treatments in our study (Figure 8). It can be seen that the difference in photosynthesis among groundwater depth treatments is caused by stomatal limitations, which differs from the research results of Xia et al. [35], mainly because of the supplemental irrigation set up in this experiment. Wang [9] discovered that the groundwater depth of 1 m significantly increased Pn and Tr compared to the groundwater depths of 2 and 3 m. A similar conclusion was obtained in this study. D1 treatment was beneficial to improving the leaf stomatal conditions and accelerated gas exchange so that the $CO_2$ content in the leaf stomata increased. The increase in photosynthetic substrate led to an increase in photosynthesis [22,34]. We also found that the D4 treatment improved soybean leaf gas exchange parameters compared to the D2 and D3 treatments. The reason for this was that the D4 treatment had a maximum irrigation volume, which improved the soil moisture conditions in the main active layer of the root system and promoted the photosynthesis of soybeans.

The DMA directly reflects the crop's photosynthetic production capacity, while the accumulation of photosynthetic products is the basis for yield formation. Different from stems and pods, the DMA in mature leaves in this study gradually decreased with the increase in groundwater depth, and the D4 treatment was at the lowest level (Figure 9). The reason is that there was no irrigation water supply during the mature period of soybeans, and with the increase in groundwater depth, the amount of groundwater recharge to soybeans was reduced, which resulted in accelerated wilting of soybean leaves. This study also pointed out that the total underground DMA and yield of soybean showed D1 > D4 > D2 > D3 under different groundwater depth treatments (Figure 9), which was consistent with the trends of the LAI, SPAD, IPAR and leaf gas exchange parameters. In the comprehensive analysis, it was believed that when the groundwater depth was 1 m (D1), the groundwater supply was sufficient, resulting in a larger soil water content in the root zone of the plant. And it was beneficial for promoting the growth of soybean leaf area and the synthesis of SPAD, increasing the absorption and utilization of solar energy, improving the condition of leaf stomatal opening and closing, accelerating gas exchange between the plant and the atmosphere and enhancing the photosynthetic production capacity of soybean. When the groundwater depth increased from 1 m (D1) to 3 m (D3), the upward recharge path of groundwater was lengthened and the amount of groundwater recharge decreased. And even if a certain level of irrigation water supplementation was obtained, the soil moisture status in the root zone of the soybeans gradually deteriorated, which in turn caused a steady reduction in the growth and development of the leaves as well as their ability to photosynthesize. Because of the excessive buried depth, capillary rising water barely touched the soil layer of plant roots when the groundwater depth reached 4 m (D4). In this instance, the maximum irrigation might also encourage the growth and development of soybean leaves, intercept more photosynthetic radiation, improve plant photosynthetic efficiency, promote the soybean DMA and provide a foundation for yield production.

## 5. Conclusions

In the two-year field experiment with an automatic groundwater control system, the effects of different groundwater depths on the ET, photosynthetic characteristics and yield of soybean were investigated. The groundwater depth affected the soybean ET and sources of ET. It was found that under D1 treatment, soybean was irrigated with seedling emergence water only once at sowing, and the entire growth period thereafter could meet the water requirements for growth and development by relying on groundwater recharge, with the largest ET and the greatest contribution of groundwater to ET. More importantly, compared

with the other three treatments, the D1 treatment promoted the growth of soybean leaf area and the generation of chlorophyll, obtained the most IPAR, improved the state of leaf stomata, increased gas exchange between the plant and the atmosphere, enhanced the photosynthetic capacity of soybean and achieved the maximum DMA and yield. With the increase in groundwater depth, the groundwater recharge gradually decreased, the growth and development of soybean leaves and their photosynthetic capacity gradually decreased and the ET and groundwater contribution to ET also gradually decreased, resulting in a synchronous reduction in DMA and yield of soybean. The D4 treatment had the smallest amount of groundwater recharge and contribution rate to ET, but its irrigation volume was the largest. The growth, development and photosynthetic performance of soybean leaves also reached a higher level and the ET enhanced to some extent, which was conducive to the DMA and yield improvement of soybean.

**Author Contributions:** Conceptualization and methodology, S.S.; formal analysis, Z.Z.; investigation, Z.Z., Z.W. and R.S.; writing—original draft preparation, Z.Z.; writing—review and editing, Z.Z., Z.C. and S.S.; visualization, Z.Z.; project administration and funding acquisition, S.S. All authors have read and agreed to the published version of the manuscript.

**Funding:** This study was supported by the Liaoning Province Applied Basic Research Program Project (2023JH2/101300123), Liaoning Province Scientific Research Funding Project (LSNFW201913) and National 13th Five Year Plan Key R&D Project (2018YFD0300301). Liaoning Province Applied Basic Research Program Project: 2022JH2/101300195.

**Data Availability Statement:** The data sets presented in this study are available within the article.

**Conflicts of Interest:** The authors declare no conflicts of interest.

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
