# Peer review of "Response of Evapotranspiration, Photosynthetic Characteristics and Yield of Soybeans to Groundwater Depth"

_agronomy, doi:10.3390/agronomy14010183_

Round 1
Reviewer 1 Report
Comments and Suggestions for Authors
Brief Summary: This is a study of the response of evapotranspiration, photosynthesis parameters, and yield to groundwater depth in soybean production. The main objective of this work was to elucidate the role of groundwater depth on soybean yield parameters and to optimize layout for soybean plantings for improved soybean yields and efficient use of water resources in shallow groundwater areas.
General concept comments: This study is important for soybean production as global water resources become more limiting to agricultural yields. Greater understanding of how groundwater depth and precipitation/irrigation patterns interact can help producers more efficiently and effectively balance the availability of these water sources to maximize water use efficiency and minimize loss of water to runoff and leaching during soybean production. Although not particularly original, the concepts investigated in this work have not been applied comprehensively to soybean, which represents a significant gap in our knowledge given the importance of soybean to global protein and calorie requirements.
The methods used in this study are appropriate, with appropriate controls in place. However, the description of the plot design and their statistical methods is lacking. In particular, the reader needs more information on the number of plants there were in each block and what number of plants was considered an experimental unit. Also, what kind of hypothesis testing did you use, and how did you test for heterogeneity in your data?
Presentation of the results was adequate, but lacking in clarity. I will give specific suggestions in my comments below.
The conclusions are consistent with the evidence and arguments presented, and they address the main question posed.
The references cited are appropriate and current.
Specific comments:
Results:
Line 213: suggest “the amount irrigation water needed increased by”
Line 216: suggest “The amount of irrigation water needed for the D1 treatment was only X, X%, X% and X%, X%, X% of that needed for the D2, D3, and D4 treatments in 2021 and 2022, respectively."
227-232: Suggest: “Compared to the D1 treatment, ET decreased significantly with the increase in groundwater depth, with the minimum ET value observed in the D3 treatment. Compared with the ET of the D1 treatment, ET of D2, D3, and D4 treatments were X%, X% and X% lower, respectively, in 2021, and X%, X% and X% lower, respectively, in 2022.
Line 247-250: “At the podding stage, when the LAI reached its peak, the LAI under the D1 treatment increased by 16.22%, compared to the D3 treatment in 2021, and increased by 19.29% and 24.88%, compared to the D2 and D3 treatments in 2022, respectively.” – Can you really say there were increases and decreases when, statistically, the only significant decrease was D3 in comparison to D1 in 2021 and between D1 and D2 and D1 and D3 in 2022?
Line 263: should be “SPAD” rather than “LAI”?
Line 263-264: I don’t understand what it means, “the D1 treatment increased by 4.94% and 5.14%, respectively, compared to the D2 and D3 treatments”. D1 values were greater than D2 and D3 values at that point in time, but an increase implies greater values over time or due to some triggering event. This wording causes confusion, and muddies your data.
Line 275-277: See my previous comment related to SPAD values
Line 284: “All” does not need to be capitalized
Line 286-299: See my previous comments about the problematic use of the term “increase” when comparing values at a given point in time.
Line 290: should be “podding stages”
310-313: See my previous comments about the problematic use of the term “increase” when comparing values at a given point in time.
Line 319: “treatments” (spelling)
Line 320-322: See my previous comments about the problematic use of the term “increase” when comparing values at a given point in time.
Line 351: “important” (spelling)
Line 384: Should be “the SPAD” (no caps for “the”)
Tables and Figures are adequate and readable.
Comments on the Quality of English Language
Line 263: should be “SPAD” rather than “LAI”?
Line 351: “important” -spelling error
Author Response
Please see the detailed response in the attachment

Reviewer 2 Report
Comments and Suggestions for Authors
Title:
Response of Evapotranspiration, Photosynthetic Characteristics and Yield of Soybeans to Groundwater Depth
The paper presents a contribution to the knowledge base of groundwater influence on soybean production. The effect of one factor, groundwater depth levels (1 m (D1), 2 m (D2), 3 m (D3), and 4 m (D4)), on soybean evapotranspiration (ET), photosynthetic characteristics and yield was investigated in field conditions in 2021 and 2022. The automatic groundwater depth control system was used and a mobile rain shelter to isolate natural precipitation. The results show the highest values in D1 than D4>D2>D3.
An abstract is understandable and written clearly. The Introduction gives the background needed to understand the study. Literature references are up-to-date and relevant. At the end of the introduction, the aim of the study was comprehensively written (a detailed remark is given below). The materials and methods section is written concisely; the methods used are described in brief, but mostly missing references for calculations. In the results and discussion section, additional explanations of the influence of water inflow on the examined parameters are necessary. The major issue in methodology that needs to be clarified is irrigation, which in this paper is a significant element in e.g. ET calculation and consequently influences the investigated physiological parameters, dry matter, yield, etc. Also, it is not clear why D4 had higher values of investigated parameters since the soil moisture was maintained in the same range on all treatments by irrigation in D2 to D4.
Figure 1 - the scheme is blurry
85 and 88 – citations should be as a reference number in square brackets, and the site in the references
102 – “the soybean planting layout” – the paper is about the critical depth of groundwater, not optimizing planting. The only investigated factor is the depth of groundwater.
129 – consider changing the subtitle to Experimental Design or Crop Management and Irrigation
139 - Table 1 – provide a reference for upper and lower limits of soil moisture control. Also, how was field capacity determined, and what is the value of FC?
139-140 – “Except for the experimental treatments, field management was in line with local farmers' practices.” – this is unclear. If the presented results are gained from the experimental field then write in detail all crop management practices. It would be interesting to see a picture of the experimental plot.
154 – provide a formula for this calculation and bulk values if used
equation 2 – how were G and SWD determined?
equation 3, 5 - – provide a reference
166-167 – provide a reference for this
196 – “blanched” – find a more appropriate term
197 – weight was determined with a balance, how was DMA determined?
366 – how is rainfall a water income if a mobile rain shelter was at the field (as stated in lines 125-126)?
369 – “more adequate irrigation” – a very unusual term, I wouldn’t say the irrigation is more or less adequate since it is based on soil moisture.
If the soil water content was maintained within the lower and upper limits based on field water capacity, in all plots, why are ET and irrigation (figure 3 and 4) different in D2 to D4 treatments since the water income is mostly from irrigation on these plots (especially on D3 and D4)? This is even more pronounced in 2022, where groundwater recharge is not significant on D2-D4 but irrigation is (figure 3). Please explain this since this is of key importance for the interpretation of the results. Also, present the soil moisture data on all D treatments.
381-382 – “further investigation revealed that this was because of the different irrigation conditions” – to what does this refer to, to Kang et al or your study (if this is the case explain how irrigation conditions are different)?
390-391, 446– “timely and sufficient water supply (supplementary irrigation)” was also on all treatments because the irrigation time was determined by soil moisture which should be the same, except in case of e.g. overmoisture by the groundwater. The same is for line 412
420 – if “D4 treatment had an adequate and timely irrigation water supply” does this mean that D2 and D3 did not have adequate and timely irrigation?
427 – “mature period” – to which growth period in Table 1 corresponds?
427 – “no irrigation water supply during the mature period of soybeans” - when was irrigation terminated on all D treatments?
430 – total underground DMA and yield – it would be much easier to follow if Figure or Table number is referred to
441 – “the soil moisture status in the root zone of the soybeans gradually deteriorated” – why did the soil moisture status deteriorate? Does this refer to growth stages or D treatments?
467 – 470 – this stands for D2-D4, not only D4.
Author Response

(The authors gave the same response as above.)

Reviewer 3 Report
Comments and Suggestions for Authors
The article «Response of Evapotranspiration, Photosynthetic Characteristics and Yield of Soybeans to Groundwater Depth» is devoted to elucidation of the physiological mechanism of the influence of different groundwater table depths on evapotranspiration, photosynthetic characteristics and soybean yields. For this purpose, a field experiment with four levels of groundwater depth (1 m (D1), 2 m (D2), 3 m (D3) and 4 m (D4)) was conducted and mathematical modeling of various indicators affecting soybean yield was carried out.
The authors have carried out a thorough and thoughtful study, which deserves publication. However, there are some questions for the authors. However, there are some questions for the authors.
I would like to note the well-written introduction, from which the aims and objectives of this study are clear.
Based on the data in Table 2, I believe that after groundwater at D1, there is a decrease in leaf area index at the Branching stage, which changes slightly at D2, D3 and D4. I.e., I do not quite agree with the authors' conclusion that the index increases after the decline, it rather fluctuates.
In Figures 3, 5, 6, and 7, it is necessary to make a legend with a colored signature on the right too.
Lines 273-277 «the IPAR under different treatments followed the following order: 274 D1>D4>D2>D3» And in 2022, the dependence is rather like this: the IPAR D1>D4>D2=D3
Line 417... photosynthetic raw materials... If I understand correctly, it means increasing the photosynthetic substrate (CO2). Please replace the word raw materials with substrate.
Author Response

(The authors gave the same response as above.)

Round 2
Reviewer 2 Report
Comments and Suggestions for Authors
The authors answered all the points of the review, and added adequate explanations regarding irrigation and soil moisture in the manuscript.
Author Response
*** Comments from Reviewer 2 : The authors answered all the points of the review, and added adequate explanations regarding irrigation and soil moisture in the manuscript.
*** Author's response : Thanks a lot for your postive comments!